# Studies on the Phosphorus-Solubilizing Ability of *Isaria cateinannulata* and Its Influence on the Growth of *Fagopyrum tataricum* Plants

**DOI:** 10.3390/plants13121694

**Published:** 2024-06-19

**Authors:** Guimin Yang, Can Liu, Lingdi Gu, Qingfu Chen, Xiaona Zhang

**Affiliations:** 1Research Center of Buckwheat Industry Technology, College of Life Science, Guizhou Normal University, Guiyang 550025, China; 222100100453@gznu.edu.cn (G.Y.); 194512020024@gznu.edu.cn (L.G.); 2School of International Education, Guizhou Normal University, Guiyang 550025, China; christine53@163.com

**Keywords:** phosphate-solubilizing fungi, *Isaria cateinannulata*, *Fagopyrum tataricum*, chlorophyll, soluble phosphorus

## Abstract

*I. cateinannulata* has been shown to promote the growth of *F. tataricum*. However, whether its growth-promoting capacity is related to its ability to solubilize phosphorus has not been reported. Therefore, in this study, we sought to assess the phosphorus-solubilizing ability of 18 strains of *I. cateinannulata* by analyzing their growth in an inorganic phosphorus culture medium. The effects of *F. tataricum* on growth and effective phosphorus content were analyzed through field experiments. The results showed that all 18 strains of *I. cateinannulata* had a phosphorus release capacity, with phosphorus solubilization ranging from 5.14 ± 0.37 mg/L to 6.21 ± 0.01 mg/L, and strain 9 exhibited the best phosphorus solubilization effect. Additionally, the field results demonstrated that *I. cateinannulata* positively influenced the growth, root length, and yield of *F. tataricum* by increasing the chlorophyll and soluble phosphorus content. This study will provide a material basis and theoretical support for investigating the interaction mechanism between *I. cateinannulata* and *F. tataricum.*

## 1. Introduction

Phosphorus (P) is one of the essential nutrient elements for plant growth and development, accounting for approximately 0.05–0.5% of plant dry weight. It plays a crucial role in various biochemical reactions, serving as an important component of cell structure and composition, and significantly influences crop yield and quality [1,2]. However, phosphorus in soil mainly exists in the form of insoluble organic or inorganic phosphates; less than 5% of available phosphorus can be directly absorbed and utilized by plants [3]. Phosphorus deficiency in soil can lead to various symptoms, such as purple spots on crop leaves and stems, reduced stomatal conductance, restricted regeneration of ribulose-1,5-bisphosphate carboxylase/oxygenase (RuBP), decreased photosynthetic rate, abnormal carbon and nitrogen metabolism, accumulation of carbohydrates, and inhibition of protein synthesis [4,5,6,7,8,9]. In severe cases, it can significantly reduce crop yield [2].

*F. tataricum* (L.) Gaertn, commonly known as *Tartary buckwheat*, is a dicotyledonous plant belonging to the family Polygonaceae, possessing characteristics such as cold and cool tolerance, poor soil tolerance, and a relatively short growth period. Moreover, it contains balanced amino acids, trace elements, dietary fibers, and other rich active substances that contribute to enhancing blood circulation; alleviating gastrointestinal stagnation; treating chronic diarrhea; and lowering blood sugar, cholesterol, triglycerides, etc. [10]. The natural pollination success rate of *F. tataricum* is relatively low, with the fruit set rate typically ranging between 15% and 35% [11,12]. In order to increase the yield of *F. tataricum*, people often apply fertilizer, with phosphate fertilizer being among the most utilized due to its capacity to enhance the filling degree and final yield of *F. tataricum* [13]. Typically, the phosphorus fertilizer applied in *F. tataricum* fields ranges from 45 kg/ha to 60 kg/ha [14,15]. However, the utilization rate during the season generally falls within a range from 10% to 25%. Unused phosphorus fertilizer can form insoluble phosphates with soil metal ions such as iron, aluminum, magnesium, and calcium in the soil [3,16,17]. Long-term and excessive use of phosphorus fertilizer can lead to phosphorus accumulation in the soil, resulting in the waste of non-renewable phosphorus mineral resources and environmental pollution [18]. Hence, exploring efficient phosphorus solubilization measures of green economy holds significant importance for addressing phosphorus deficiency in crops and non-point source pollution, maintaining ecological balance, etc.

Phosphate-solubilizing microorganisms (PSMs) have attracted much attention due to their ability to convert insoluble bound phosphorus into soluble free phosphorus that can be absorbed and utilized by plants, as well as their ability to interact with plants and regulate the effectiveness of plant rhizospheric phosphorus. Phosphate-solubilizing fungi, in particular, have emerged as a focal point for research due to their high phosphorus solubilization efficiency, strong adaptability, and high genetic stability [19]. It has been found that fungi generally have several times higher phosphorus solubilization ability than bacteria [20,21]. Furthermore, many bacteria lose their phosphorus solubilization ability during subculture, while fungi consistently maintain strong phosphorus solubilization activity and genetic characteristics [22,23,24]. Currently identified phosphate-solubilizing fungi include species such as *Aspergillus niger* [25], *A. aculeatus* [26] from the genus *Aspergillus*, *Penicillium oxalicum* [27], *P. pinophilum* [28] from the genus *Penicillium* [29,30,31], *Fusarium oxysporum* [32] from the genus *Trichoderma* [33,34], and *Paecilomyces lilacinus* [35] from the genus *Paecilomyces* [36,37].

*I. cateniannulata*, also known as *Paecilomyces cateniannulatus* or *Cordyceps cateniannulata*, belongs to the subphylum Entomophthoromycotina, class Zygomycetes, order Entomophthorales, and genus Cordyceps. It serves as an entomopathogenic fungus that is capable of controlling pests and diseases, while also promoting plant growth [38,39]. At present, it is found to enhance the growth of Tartary buckwheat seeds [38,40], tobacco [41,42], tomatoes [43], and other plants through endophytic colonization. However, it has not been reported whether its promotion of plant growth is related to its phosphorus-solubilizing ability.

Therefore, this study aimed to explore the phosphorus-dissolving ability of 18 strains of *I. cateniannulata* by analyzing their growth in an inorganic phosphate culture medium. Through field experiments, the influence of these strains on the growth of *F. tataricum* was assessed. The study is expected to provide the material and theoretical basis for the interaction mechanism of *I. cateniannulata* and *F. tataricum* and the research of *I. cateniannulata* as phosphate-solubilizing fungi.

## 2. Materials and Methods

### 2.1. Solid Medium Preparation and Determination of Phosphorus-Solubilization Index of Strains

Solid medium formulation: 10 g glucose, 0.5 g ammonium sulfate (Shanghai Aladdin Biochemical Technology Co., Shanghai, China), 0.3 g sodium chloride, 0.3 g potassium chloride (Tianjin Windship Chemical Reagent Technology Co., Tianjin, China), 0.03 g ferrous sulfate (Sinopharm Chemical Reagent Co., Shanghai, China), 0.5 g tricalcium phosphate (Chongqing Jiangchuan Chemical Co., Chongqing, China), 0.03 g manganese (II) sulfate tetrahydrate (Shanghai Eon Chemical Technology Co., Shanghai, China), 0.3 g zinc sulfate heptahydrate (Beijing Soleberg Technology Co., Beijing, China), 20 g Agar-agar. The solution was prepared by dissolving the mixture in 1000 mL of purified water [34]. The solution was transferred into 500 mL triangular flasks, with each flask not exceeding 300 mL. High-pressure sterilization was performed at 121 °C for 15 min. Afterward, the solution was transferred into culture dishes of 9 cm diameter, with each dish containing 25 of the medium. These dishes were then ready for use.

The strains of *I. cateniannulata* (Nos. 1–18, isolated from soil in buckwheat production area in Guizhou Province, China, currently preserved in the Research Center of Buckwheat Engineering and Technology, School of Life Sciences, Guizhou Normal University) were cultured on solid culture medium. After 15 days, a spore suspension with a concentration of 1 × 10^10^ spores/mL was prepared using sterile distilled water. Holes with a diameter of 0.2 cm were made in the center of the solid culture medium, and a volume of 10 μL of the spore suspension was inoculated into each circular hole, using a pipette gun (operated on super-clean worktable). The cultivation was conducted in an artificial climate box (manufactured by Ningbo Yanghui Instrument Co., Ltd., Ningbo, China) at a temperature of 25 °C, humidity of 70%, light intensity of 2000 lx, and light/dark cycle of 12:12. Observations and photographs were taken every 24 h. Measurements of colony diameter (D) and transparent zone diameter (d) were taken at multiple perpendicular directions on the 6th, 8th, and 10th days. The ratio of transparent zone diameter to colony diameter (D/d) was calculated as the phosphorus-solubilization index (PSI).

### 2.2. Preparation of Liquid Medium and Determination of Soluble Phosphorus Content of Strains

Liquid culture medium preparation: Agar powder was removed from 1.1 solid culture medium, and the remaining components were mixed in a 1000 mL beaker. Distilled water was added to reach a total volume of 1000 mL, and the mixture was dispensed into a 150 mL triangle bottle, with each flask containing 50 mL. The medium was then autoclaved at 121 °C for 15 min and allowed to cool below 25 °C. Next, 1 mL of fungal spore suspension with a concentration of 1 × 10^10^ spores/mL was added to each triangle bottle. The triangle bottle was cultivated continuously for 5 days at A temperature of 25 °C, humidity of 70%, light intensity of 2000 lx, and light/dark cycle of 12:12, with a rotation speed of 150 r/min. The pH of the medium was measured using a pH meter (Nanjing Sutest Instrument Co., Ltd., Nanjing, China) and then stored for future use.

Potassium phosphate monobasic (analytical reagent) (Chongqing Jiangchuan Chemical Co., Ltd., Chongqing, China) was accurately weighed to 0.4398 g, dried to constant weight, and then dissolved in distilled water to make a final volume of 1000 mL. The solution was mixed well, and 10 mL of this solution was taken and diluted to 100 mL with distilled water to prepare a 5 mg/L phosphate standard solution. Next, 0, 2, 4, 6, 8, 10, 12, and 14 mL of the phosphate standard solution were pipetted into separate 50 mL volumetric flasks. Dinitrophenol indicator (Shanghai Yien Chemical Technology Co., Ltd., Shanghai, China) was added (400 µL), followed by titration with 100 g/L sodium carbonate solution until the color of the solution just turned faint yellow. Then, 5 mL of molybdenum antimony color reagent (Xiamen Haibiao Technology Co., Ltd., Xiamen, China) was added, mixed well, and diluted to volume with distilled water. The solution was placed at room temperature above 15 °C for 30 min. The absorbance at a wavelength of 700 nm was measured using a UV–visible spectrophotometer (Beijing Puxi General Instrument Co., Ltd., Beijing, China), with absorbance as the *y*-axis and phosphate concentration as the *x*-axis, to plot the phosphate standard curve 1.

### 2.3. Determination of Phosphorus Solubilization Amount by I. cateinannulata

The molybdenum antimony colorimetric method [44] was used to determine the amount of phosphorus solubilization by *I. cateinannulata* strains: The 25 mL culture broth after 5 days of cultivation was centrifuged at 12,000 rpm for 5 min. Then, 4 mL of the supernatant was transferred to a 50 mL volumetric flask, followed by the addition of 26 mL of distilled water. Next, 400 µL of dinitrophenol indicator was added, and the solution was adjusted to a faint yellow color using a 100 g/L sodium hydroxide solution. Subsequently, 5 mL of molybdenum antimony color reagent was added, and the solution was shaken and diluted with distilled water to the mark. It was left to stand at a temperature above 15 °C for 30 min. The sterile liquid culture medium treated under the same conditions was used as a reference to adjust the zero point. The absorbance of the chromogenic solution was measured at a wavelength of 700 nm, using a UV–visible spectrophotometer, and the corresponding soluble phosphorus content was determined from the standard curve

### 2.4. Field Test Ring I. cateinannulata Strain on the Growth of F. tataricum

#### 2.4.1. *I. cateinannulata* Roots Irrigation for *F. tataricum*

*F. tataricum* cultivation: From August to December 2023, small-scale experiments were conducted in the field (at the Experimental Base of Buckwheat Industry Technology Center, Guizhou Normal University, Anshun City, Guizhou Province, located at 106°7′7″ E, 26°5′9″ N, and an altitude of 1397 m). The experiments were conducted in plots with consistent water and fertilizer conditions. Variety rice *F. tataricum* 13 was selected as the experimental material. Each treatment was allocated to one plot, with each plot consisting of 3 rows and each row containing 80 plants. Each treatment was repeated 3 times, totaling 3 m^2^. A 1 m interval separated the experimental and control groups, and the management of the field was identical.

Rhizosphere inoculation experiment with *I. cateinannulata*: During the *F. tataricum* seedling stage (at 15 days of planting), spore suspensions of the strains exhibiting the highest phosphorous solubilizing capacity were selected for root irrigation. The irrigation was conducted for three consecutive days, with one application before sunrise and one application after sunset each day. The dosage for each application was 1.6 × 10^9^ spores/m^2^, totaling 6 times.

#### 2.4.2. Determination of Growth Index of *F. tataricum*

After 15 d, 30 d, and 80 d of root irrigation, the *F. tataricum* control group and experimental group were collected, with 10 plants per row and 30 plants per each treatment. They were then brought back to the laboratory to measure the plant height, stem diameter, number of branches, number of stems, length of the main root, and leaf area of *F. tataricum*.

#### 2.4.3. Chlorophyll Determination

Fresh leaves of *F. tataricum* plants were taken, and the surface of the tissues were cleaned and cut into pieces (removing the midrib). Then, 0.1 g of the sample was weighed and placed into a 25 mL test tube. Subsequently, 15 mL of 95% ethanol was added to the test tube to completely immerse the sample in the ethanol solution. It was placed in darkness for 24 h to soak in order to turn into chlorophyll pigment extract. The extract was poured into a colorimetric cup, and 95% ethanol was used as a blank control. The absorbance was measured at wavelengths of 665 nm, 649 nm, and 470 nm [45]. The concentrations of chlorophyll a (Ca) and chlorophyll b (Cb) were calculated using the following formula (where A represents absorbance), as well as the content of each pigment per unit fresh weight in the tissue (unit mg/g):Ca = 13.95 × A_665_ − 6.88 × A_649_
Cb = 24.96 × A_649_ − 7.32 × A_665_
The content of chloroplast pigments = (pigment concentration × extraction liquid volume)/sample fresh weight

#### 2.4.4. Determination of Soluble Phosphorus Content in *F. tataricum* Plants

A total of 25 g of ammonium molybdate (provided by Chongqing Jiangchuan Chemical Co., Ltd., Chongqing, China) was weighed and dissolved in distilled water to make up a volume of 1000 mL to obtain a 2.5% ammonium molybdate solution. Then, 10 g of ascorbic acid (provided by Tianjin Zhiyuan Chemical Reagent Co., Ltd., Tianjin, China) was weighed and dissolved in 100 mL of distilled water to obtain a 10% ascorbic acid solution (due to its susceptibility to oxidation, it is prepared and used immediately). A mixture of distilled water, 6 mol/L phosphoric acid, 2.5% ammonium molybdate, and 10% ascorbic acid in a 2:1:1:1 volumetric ratio was prepared and stored in a brown bottle, serving as the phosphorus reagent (the liquid should not be used if it turns brown-yellow).

The phosphorus standard was prepared by taking 0, 0.2, 0.4, 0.6, 0.8, 1.0, and 1.4 mL from the 2.2 and placing them in 10 mL centrifuge tubes. To each tube, 3 mL of phosphorus reagent was added and made up to 6 mL with distilled water. The tubes were incubated in a 45 °C water bath for 25 min. The absorbance was measured at a wavelength of 660 nm, using an ELISA reader. The phosphorus standard curve 2 was plotted, with absorbance as the *y*-axis and phosphorus concentration as the *x*-axis.

Following the phosphomolybdenum blue colorimetric method [46], 0.1 g of fresh buckwheat roots, stems, leaves, and mature seeds was placed in a mortar. Then, 1.5 mL of distilled water was added separately, and the mixture was grinded into a homogenate and placed in a 2 mL centrifuge tube. The slurry was transferred to 2 mL centrifuge tubes and centrifuged at 6000 r/min for 15 min. From the supernatant, 0.2 mL was taken and mixed with 3 mL of phosphorus reagent. After dilution to 6 mL with distilled water and complete mixing, the tubes were incubated in a 45 °C water bath for 25 min. Afterward, 0.2 mL was taken to measure the absorbance at a wavelength of 660 nm [47]. The soluble phosphorus content in the samples was calculated based on the standard curve 2 (unit mg/g).
Sample solubility phosphorus content = Mx × V/V_1_ × W

In the formula: Mx is the phosphorus content (mg/L) in standard curve 2, V is the volume of sample extraction solution (mL), V_1_ is the volume of the sample used for measurement (mL), and W is the fresh weight of the sample (g).

### 2.5. Data Processing and Analysis

Excel 2022 was used for basic data statistical analysis and column chart/line chart creation of parameters such as the phosphorus index, phosphorus solubilization amount, and plant height. SPSS 24.0 or above was used for the statistical analysis of the data. The independent samples t-test method was applied to analyze the significant differences between the two groups of data at a significance level of 0.05. Tukey’s method was used to analyze the significant differences among multiple groups of data at a significance level of 0.05. Adobe Illustrator 2023 was used for image processing.

## 3. Results and Discussion

### 3.1. Analysis of Phosphorus-Solubilization Ability among Different Strains of I. cateinannulata

Based on observations over a period of 10 consecutive days, 17 strains of *I. cateinannulata* showed the formation of clear zones on solid culture medium (Figure 1c,d). As the strains grew, the clear zones gradually increased in size (Figure 1e,f), reaching their maximum size on the 10th day (Figure 1g,f). This increase occurs because phosphorus-solubilization fungi produce a large amount of organic acid substances during their growth process, such as oxalic acid, citric acid, glucose succinic acid, malic acid, acetic acid, and so on [48,49]. These substances can decompose calcium phosphate in the culture medium, resulting in the formation of a specific transparent zone around the fungal body. It is speculated that the growth of *I. cateinannulata* produces organic acid substances capable of decomposing calcium phosphate, but the specific types need further verification. The size of the transparent zone is mainly related to the phosphorus-solubilization index and phosphorus-solubilization ability of the phosphorus-dissolving fungus, and the phosphorus-solubilization index can be used as a preliminary indicator to characterize the phosphorus-solubilization ability of the phosphorus-solubilization fungus [50]. As shown in Table 1, the phosphorus-solubilization indices of the 17 fungal strains varied, consistent with Figure 1. These differences among different strains are influenced by the phosphate-solubilization ability, genetic characteristics, environmental factors, and so on [51]. On the 6th and 8th days, strain 2 had the highest phosphate-solubilizing index, which was significantly higher than that of other strains, except strain 2, 3, 6, 7, 11, and 17 (*p* < 0.05). On the 10th day, strain 7 had the highest pi; it was significantly higher than other strains, except 1 and 2 (*p* < 0.05). Strains 2, 6, 11, 14, and 18 showed a trend of initially increasing and then decreasing in terms of their phosphorus-solubilization index as time increased. These strains reached a maximum value on the 8th day, whereas, for the other strains, the phosphorus-solubilization index gradually increased to the maximum value on the 10th day. This occurs because certain phosphorus-solubilizing fungi preferentially utilize the solubilized phosphorus after decomposing the insoluble phosphorus [52], temporarily suspending or reducing the decomposition of insoluble phosphorus, resulting in an initial increase and subsequent decrease in the phosphorus-solubilization index. Additionally, it is possible that some strains have a single phosphorus-solubilization mechanism, while others have multiple mechanisms acting together, resulting in differences in their effectiveness [24,53]. The differences in phosphorus-solubilization ability among different strains of *I. cateinannulata* may be related to all of these reasons, which are currently under-validated. The plate “transparent zone” method is suitable only for the initial screening of phosphorus-solubilizing microorganisms due to its convenience and fast speed, but it lacks full reliability. Strain 15 did not exhibit a transparent zone (Figure 1a,b) and did not have a phosphorus-solubilization index (Table 1). Possible explanations include insufficient acidity produced by the strain, resulting in an in an unclear dissolution zone, or vigorous growth of the mycelium obscuring the small dissolution zone, among other reasons [54]. Further research is needed.

Currently, the common methods for assessing the ability of phosphate-solubilizing fungi to solubilize phosphorus include a size measurement of the clear zone in a solid culture medium, the determination of the soluble phosphorus content in liquid culture medium, and an evaluation of available phosphorus content in soil–sand culture medium. Tian Xiang et al. screened a highly efficient phosphate-solubilizing strain using the phosphorus-solubilizing circle method and liquid culture method [55]; Sun Yaqin et al. obtained the FLP36 strain with high degradation ability for calcium phytate through the phosphorus-solubilizing circle method and liquid culture method [56]; and Lin Yuzha et al. evaluated the phosphorus-solubilizing ability of the P6 strain using three methods simultaneously [57]. In this study, the first two methods were employed to demonstrate, for the first time, the phosphorus-solubilizing capability of *I. cateinannulata*, with the observed phosphorus-solubilization range consistent with that of most phosphorus-releasing fungi [58,59].

### 3.2. Analysis of Soluble Phosphorus Content and pH in Different Strains of I. cateinannulata

The cultivation of the 18 strains of *I. cateinannulata* revealed a soluble phosphorus content ranging from 5.14 ± 0.37 mg/L to 6.21 ± 0.01 mg/L. Notable variations were observed among strains, with strain 9 and strain 4 demonstrating the highest and lowest soluble phosphorus content, respectively. Strain 9 was significantly higher than strains 4 and 11 (*p* < 0.05) (Figure 2a). This difference in phosphorus-solubilization ability among different strains can be attributed to variations in phosphorus mobilization mechanisms and efficiency. Despite not producing a transparent zone in solid culture medium, strain 15 exhibited a notable phosphorus-solubilization ability in the quantitative determination of soluble phosphorus content. This can be attributed to the strain’s enhanced ability to decompose and absorb phosphorus in the liquid culture system, potentially linked to the strain’s environmental adaptability [60]; however, the specific reasons remain to be explored. Figure 2b shows the standard curve, with a linear regression equation of y = 0.0916x + 0.0003 and an R^2^ value of 0.9992. The results indicate a strong linear correlation between the two variables.

After 10 days of continuous cultivation, the pH of 18 strains of *I. cateinannulata* showed varying degrees of decline (Figure 2c). The pH ranged from 3.77 ± 0.20 to 4.75 ± 0.04. This is because 1/7 of the metabolites produced by *I. cateinannulata* are acidic organic compounds. Different strains secrete different types, amounts, and concentrations of organic acids during their growth, leading to different degrees of pH decline in the culture medium [61,62].

Variations in phosphorus-solubilizing ability among different strains of phosphorus-releasing fungi mainly stem from differences in their phosphorus release mechanisms. The *Penicillium oxalicum* y2 has been found to degrade phosphorus through phosphatase action without producing organic acids in the culture medium [63]. Conversely, strain HB1 of *Penicillium oxalicum* releases phosphorus through the secretion of H^+^ ions [33], while strain NJDL-03 solubilizes calcium phosphate by producing organic acids, mainly gluconic acid and oxalic acid [64]. In this study, differences in phosphorus-solubilizing ability among different strains of *I. cateinannulata* are attributed to the types and levels of organic acids secreted by the fungi. It has been found that *I. cateinannulata* secretes 93 kinds of acid substances, such as acetic acid and propionic acid, during the colonization process [38]. The large amounts of organic acids can lower the environmental pH, providing optimal conditions for phosphorus-releasing fungi to release phosphorus [65]. This acidic environment facilitates the conversion of insoluble phosphate into plant-utilizable free phosphate [66]. Lin Qimei et al. reported that phosphorus-solubilizing fungi maintain the pH of the fungal broth at around 3.0 during their growth and reproduction by producing organic acids and protons, leading to the structural destruction of phosphate rocks [67]. In this study, the pH of *I. cateinannulata* was slightly higher than that of other phosphorus-solubilizing fungi [66], which may be related to the concentration, types, and content of the secreted organic acids [68,69].

### 3.3. The Effect of I. cateinannulata on the Soluble Phosphorus Content in F. tataricum Plants

The application of *I. cateinannulata* irrigation can significantly increase the soluble phosphorus content in the roots, stems, and leaves of *F. tataricum* seedlings during the seedling stage (*p* < 0.05). These increases are 1.38 times, 1.15 times, and 1.28 times higher compared to the control group, as shown in Figure 3a. This is because phosphate-solubilizing fungi increase the soluble phosphorus content in the soil through various phosphorus-solubilizing mechanisms, such as organic acid secretion and phosphatase production, thus facilitating the absorption of soluble phosphorus by crops and promoting the growth of seedling roots and stems [67,70,71].

During the filling stages, the load on the stems of *I. cateinannulata* increases, and phosphorus can increase stem toughness and improve lodging resistance. At this time, the transport capacity of phosphorus is much greater than in other periods. At the same time, *I. cateinannulata* undergoes blooming and fruiting, requiring phosphorus for tillering and earing promotion. In this study, the soluble phosphorus content and total soluble phosphorus content in the roots, stems, and leaves of bitter buckwheat during the grouting period were significantly higher than those in the control group (*p* < 0.05), with increases of 1.38 times, 1.91 times, 1.27 times, and 1.33 times, respectively (Figure 3b). This indicates that the use of *I. cateinannulata* can increase the phosphorus content in *F. tataricum* during the tillering period, ensuring a balanced nutrient status and fulfilling plant-growth requirements.

The soluble phosphorus content in the seeds during the filling stages showed no significant difference compared to the control group (*p* > 0.05), as depicted in Figure 3b. This phenomenon arises from the predominant focus of *F. tataricum* growth and development on the seed maturation during the tillering period. Phosphorus plays a role in coordinating the source–sink relationship in plants, promoting the translocation of photosynthetic products to the grains [72], thereby increasing grain number and weight [73]. Consequently, phosphorus did not accumulate in large quantities in the seeds during the tillering period, as shown in Figure 3b. The soluble phosphorus content of mature *F. tataricum* seeds (at 80 days after roots irrigation) was significantly higher than that of the control group (*p* < 0.05), at 238.64 ± 1.84 mg/g; thus, it is 1.42 times higher than that of the control group (Figure 3c). This elevation stems from the diminished need for phosphorus in coordinating growth and development after seed maturation, resulting in its accumulation in the seeds and a significantly higher phosphorus content in the experimental group relative to the control group. The standard curve in Figure 3d follows a linear regression equation of y = 0.2702 + 0.0312, with an R^2^ value of 0.9984. The results indicate a strong linear correlation between the two variables.

Phosphate-solubilizing fungi enhance the conversion of available phosphorus in the soil, increasing the soluble phosphorus content in crops and thereby boosting crop growth and productivity. These fungi secrete organic acids that activate insoluble phosphorus in the soil, increasing the content of available phosphorus [74]. PSWY, a phosphate-solubilizing fungi, improves the available phosphorus content in the soil, promotes the development of pepper roots, increases aboveground dry weight, and significantly enhances yield [75]. Similarly, the phosphate-solubilizing fungi G17 and WJ27 promote the growth of Masson pine by increasing soil nutrients and the content of P, N, and K in seedlings [76]. In this study, *I. cateinannulata* produced a large amount of fine mycelium in the soil, reaching areas beyond the reach of root growth, expanding the root surface area, and increasing the rhizosphere coverage [77,78]. This allowed plant roots to access previously unavailable phosphorus, which was subsequently decomposed and made available for plant absorption, thereby increasing the soluble phosphorus content in crops.

### 3.4. Effect of Rhizosphere Irrigation with I. cateinannulata on Chlorophyll Content of F. tataricum

*I. cateinannulata* irrigation significantly increased the content of chlorophyll a, chlorophyll b, and total chlorophyll in the seedling stage of *F. tataricum* compared to the control group (*p* < 0.05). The respective values were 777.01 ± 9.99 mg/g, 707.54 ± 13.64 mg/g, and 1470.48 ± 21.76 mg/g, which were 1.10 times, 1.61 times, and 1.58 times higher than the control group (Figure 4a). This can be attributed to the increase in phosphorus content in *F. tataricum* induced by the *I. cateinannulata* increase, where the surplus phosphorus accelerated the synthesis of substances such as DNA, RNA, ATP, etc. [79,80]. Consequently, this leads to chlorophyll accumulation and an augmentation in the chlorophyll content within the seedling leaves. In contrast, there is no significant difference in the content of chlorophyll a, chlorophyll b, and total chlorophyll in the grain filling stage of *F. tataricum* compared to the control group (*p* > 0.05) (Figure 4b). This observation could be attributed to the increase in soil-available phosphorus concentration over time, inducing a state of high phosphorus level in the plants. Moreover, thickening or yellowing of leaves can inhibit chlorophyll synthesis and reduce the rate of chlorophyll synthesis [81].

Phosphate-solubilizing fungi influence the chlorophyll content of crops by increasing the available phosphorus. These fungi elevate crop phosphorus levels, promote the synthesis of ATP and NADPH, and enhance chlorophyll content in Chinese fir seedlings, thus facilitating plant growth [82]. Additionally, Song Wenchao et al. observed that phosphorus in the soil can cause fluctuations in chlorophyll-a production in red-leaf grass, thereby increasing the chlorophyll content [83]. In this study, the *I. cateinannulata* was found to enhance the chlorophyll content of *F. tataricum* leaves, possibly through gene expression involved in photosynthetic pathways [84], and it also increased the nitrogen and phosphorus content in the leaves [85]. However, further research is needed to elucidate the underlying mechanisms.

### 3.5. The Effect of I. cateinannulata on the Growth of F. tataricum at Different Stages

After root irrigation with *I. cateinannulata*, the stem diameter, stem node number, branch number, and leaf area of *F. tataricum* at the seedling stage (Figure 5a), filling stage (Figure 5b), and mature stage (Figure 5c) were not significantly different from those of the control group (*p* > 0.05). However, the plant height of *F. tataricum* was significantly higher than that of the control group, i.e., 1.17 times, 1.07 times, and 1.09 times higher than the control group, respectively (*p* < 0.05) (Table 2). This phenomenon can be attributed to the promotion of *F. tataricum* growth via root irrigation with *I. cateinannulata* [38,40]. Additionally, the main root length in the experimental group significantly surpassed that in the control group, being 1.24 times, 1.30 times, and 1.18 times higher than the control group, respectively (*p* < 0.05) (Table 2). This result correlated with the available phosphorus content in the environment. When plants are grown in an environment with sufficient phosphorus, it can promote the elongation of main root cells [86,87], and vice versa. The ability of roots to absorb phosphorus is a major limiting factor in plant growth, and factors such as root morphology, physiology, biochemistry, and symbiotic characteristics are related to the efficiency of plant phosphorus absorption. This indirectly indicates that *I. cateinannulata* can promote an increase in the concentration of available phosphorus in the soil; induce main root cell division, proliferation, and elongation; enhance the absorption capacity of *F. tataricum* roots for available phosphorus; and increase plant height.

The seed filling and final yield [88,89] typically correlate with the crop-available phosphorus content. Phosphorus deficiency during the growth stage may cause a metabolic imbalance and an abnormal accumulation of dry matter, inhibiting protein synthesis [88]. In this study, the seed yield per square meter at the maturity stage of *F. tataricum* (Figure 5f) in the experimental group was 2.26 times higher than that of the control group (Table 2). This is due to the fact that *I. cateinannulata* can enhance the absorption of soluble phosphorus by *F. tataricum*, and sufficient phosphorus can promote tillering, spike production, and yield increase. However, the specific mechanism still needs further investigation.

## 4. Conclusions

This study is the first to demonstrate the phosphate-solubilizing ability of *I. cateinannulata*, with phosphorus solubilization ranging from 6.21 ± 0.01 mg/L to 5.14 ± 0.37 mg/L. *I. cateinannulata* promotes *F. tataricum* growth and yield by lowering environmental pH and increasing chlorophyll and soluble phosphate content in leaves. This research provides a material and theoretical basis for developing *I. cateinannulata* as a phosphate-solubilizing fungal fertilizer and investigating its phosphate-solubilizing mechanism. 

## Figures and Tables

**Figure 1 plants-13-01694-f001:**
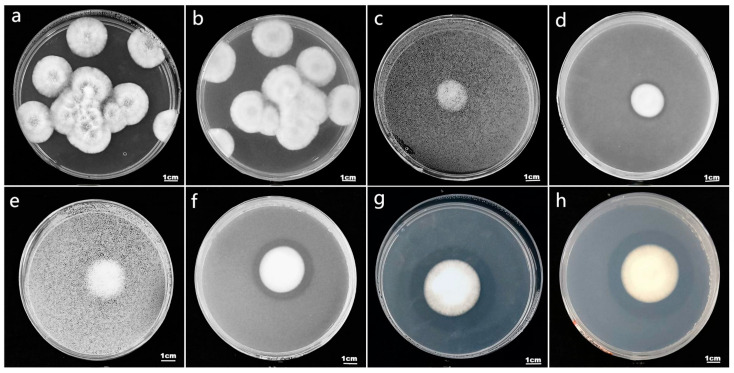
The colony characteristics and transparent zone morphology of strain 15 and strain 9. (**a**) The front view of strain 15 on the 10th day shows the colony characteristics. (**b**) The back view of strain 15 on the 10th day shows the colony characteristics. (**c**) The front view of strain 9 on the 6th day shows the colony characteristics. (**d**) The back view of strain 9 on the 6th day shows the colony characteristics. (**e**) The front view of strain 9 on the 6th day shows the colony characteristics. (**f**) The back view of strain 9 on the 8th day shows the colony characteristics. (**g**) The front view of strain 9 on the 10th day shows the colony characteristics. (**h**) The back view of strain 9 on the 10th day shows the colony characteristics.

**Figure 2 plants-13-01694-f002:**
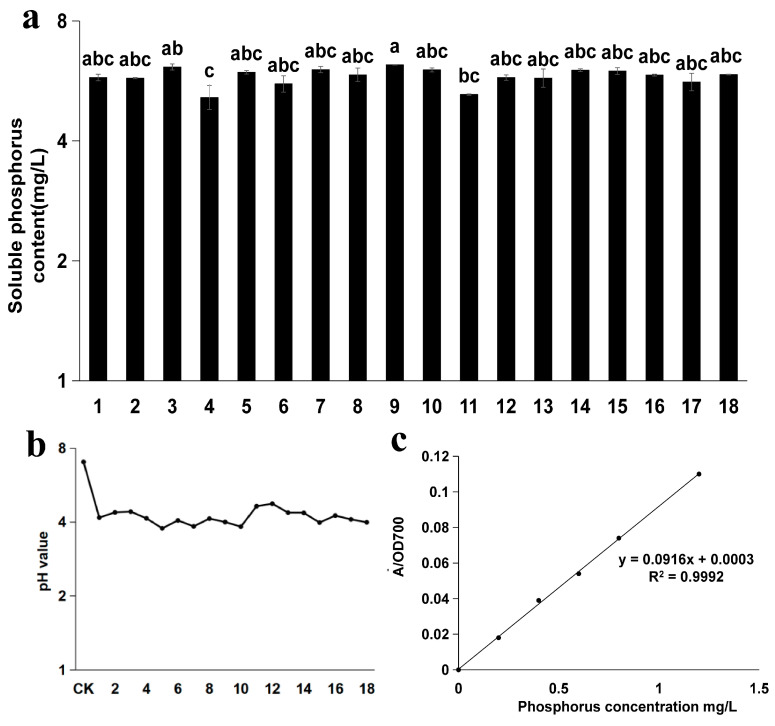
Phosphorus content and pH of 18 strains of *Isaria cateinannulata*. (**a**) Soluble phosphorus content (mg/L) in each bacterial solution after 5 days of cultivation. (**b**) Standard curve for measuring soluble phosphorus content in the bacterial solution. (**c**) pH value of each bacterial solution after 5 days of cultivation.

**Figure 3 plants-13-01694-f003:**
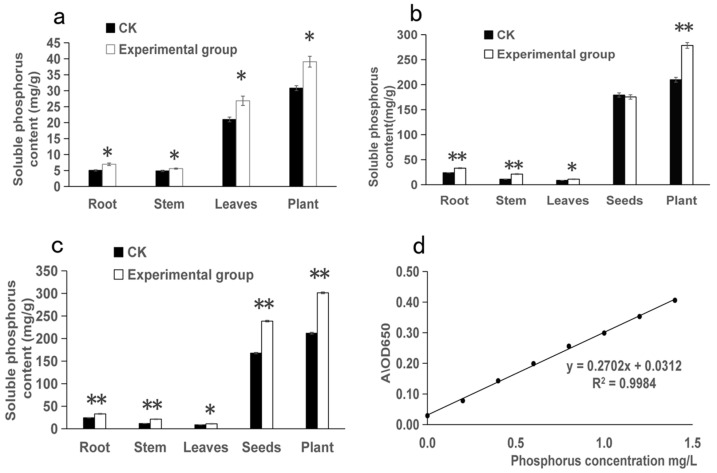
The effect of *Isaria cateinannulata* inoculation on the soluble phosphorus content of different stages of *Fagopyrum tataricum* under ring-chain rod bundle spore irrigation in the CK (control group) and experimental group (the treatment group after *I. cateinannulata* root irrigation). (**a**) The soluble phosphorus content (mg/g) in various parts of *Fagopyrum tataricum* during the seedling stage (15 days after root irrigation); (**b**) the soluble phosphorus content (mg/g) in various parts of *Fagopyrum tataricum* during the filling stage (30 days after root irrigation); (**c**) the soluble phosphorus content (mg/g) in various parts of *Fagopyrum tataricum* during the mature stage (80 days after root irrigation); and (**d**) the standard curve for measuring the soluble phosphorus content in different parts of *Fagopyrum tataricum*. ‘*’ indicates that the difference between the experimental group and the control group is significant, and ‘**’ indicates that the difference between the experimental group and the control group is extremely significant.

**Figure 4 plants-13-01694-f004:**
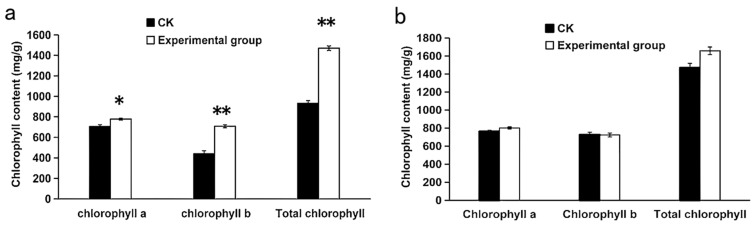
The effect of *Isaria cateinannulata* root irrigation on the chlorophyll content of *Fagopyrum tataricum* leaves at different stages. (**a**) Chlorophyll content in leaves of *Fagopyrum tataricum* seedlings (15 days after root irrigation) is measured in mg/g. (**b**) Chlorophyll content in leaves of *Fagopyrum tataricum* during the filling stage (30 days after root irrigation) is measured in mg/g. ‘*’ indicates that the difference between the experimental group and the control group is significant, and ‘**’ indicates that the difference between the experimental group and the control group is extremely significant.

**Figure 5 plants-13-01694-f005:**
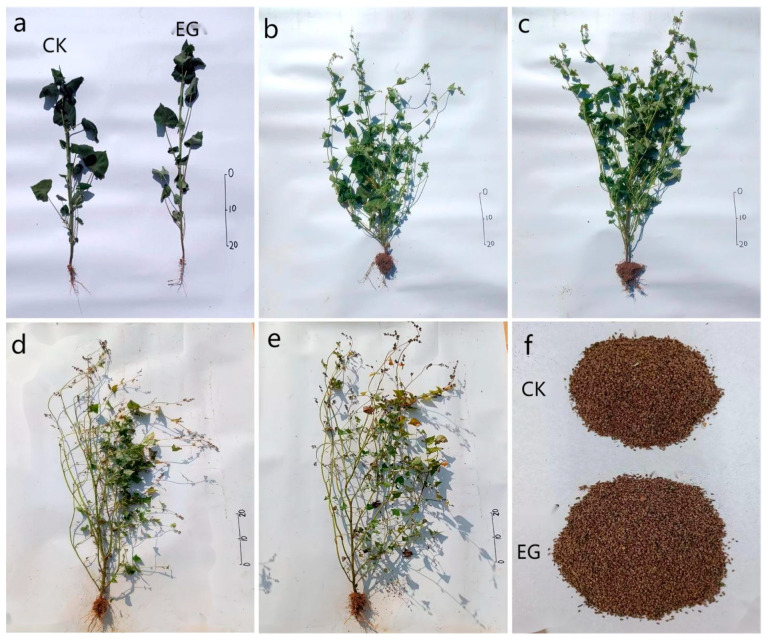
The *Fagopyrum tataricum* and yield at different stages of roots irrigation with *Isaria cateinannulata.* (**a**) Seedling-stage *Fagopyrum tataricum*. (**b**) Control-group *Fagopyrum tataricum* in the filling stage. (**c**) Experimental-group *Fagopyrum tataricum* in the filling stage. (**d**) Control-group *Fagopyrum tataricum* in the mature stage. (**e**) Experimental-group *Fagopyrum tataricum* in the mature stage. (**f**) Control-group and experimental-group 1 m^2^ yield.

**Table 1 plants-13-01694-t001:** Phosphorus-solubilization index of 18 strains of *Isaria cateinannulata*.

Code in Paper	PSI		
	6 d	8 d	10 d
1	1.14 ± 0.05 b	1.21 ± 0.00 cd	1.52 ± 0.08 ab
2	1.58 ± 0.02 a	1.84 ± 0.11 a	1.48 ± 0.24 ab
3	1.27 ± 0.27 ab	1.24 ± 0.24 bcd	1.27 ± 0.27 bc
4	1.11 ± 0.03 b	1.12 ± 0.06 d	1.24 ± 0.08 bc
5	1.05 ± 0.02 b	1.11 ± 0.07 d	1.21 ± 0.11 bc
6	1.53 ± 0.03 a	1.56 ± 0.11 abc	1.00 ± 0.00 bc
7	1.32 ± 0.11 ab	1.58 ± 0.11 ab	1.86 ± 0.07 a
8	1.14 ± 0.04 b	1.18 ± 0.07 cd	1.19 ± 0.05 bc
9	1.07 ± 0.04 b	1.16 ± 0.09 d	1.25 ± 0.13 bc
10	1.03 ± 0.03 b	1.08 ± 0.05 d	1.13 ± 0.09 bc
11	1.56 ± 0.28 a	1.60 ± 0.30 ab	1.33 ± 0.33 bc
12	1.12 ± 0.04 b	1.20 ± 0.07 cd	1.27 ± 0.10 bc
13	1.02 ± 0.02 b	1.05 ± 0.02 d	1.07 ± 0.04 bc
14	1.08 ± 0.02 b	1.14 ± 0.05 d	1.13 ± 0.05 bc
15	0.00 ± 0.00 b	0.00 ± 0.00 e	0.00 ± 0.00 d
16	1.11 ± 0.00 b	1.19 ± 0.01 cd	1.29 ± 0.04 bc
17	1.56 ± 0.05 a	1.70 ± 0.03 a	1.84 ± 0.10 a
18	1.11 ± 0.03 b	1.20 ± 0.10 cd	1.18 ± 0.09 bc

Note: The values in the text represent the mean ± standard error. Lowercase letters indicate no significant difference at the 0.05 level, while different letters indicate significant differences.

**Table 2 plants-13-01694-t002:** The effects of root irrigation with *Isaria cateinannulata* at different stages on the growth of *Fagopyrum tataricum*.

Stage		Height (cm)	Stem Thick (mm)	Number of Stems	Number of Branches	Taproot Long (cm)	Leaf Area (cm^2^)	*F. tataricum* Yield (kg/m^2^)
Seeding stage	CK	48.23 ± 1.39	5.48 ± 0.14	10.00 ± 0.23	8.78 ± 0.40	87.06 ± 3.65	522.99 ± 20.09	—
	EG	56.28 ± 0.56 *	5.36 ± 0.17	10.33 ± 0.24	8.78 ± 0.40	108.06 ± 1.9 *	610.96 ± 30.70	—
Filling stage	CK	99.65 ± 1.29	6.81 ± 0.28	18.56 ± 0.18	10.44 ± 0.18	89.82 ± 3.82	3228.45 ± 153.81	—
	EG	106.23 ± 1.58 *	6.96 ± 0.19	19.00 ± 0.33	10.67 ± 0.24	115.81 ± 3.66 **	3429.11 ± 184.94	—
Maturation stage	CK	104.16 ± 0.72	6.94 ± 0.29	20.00 ± 0.24	11.11 ± 0.31	98.45 ± 1.14	1340.7 ± 81.76	0.35 ± 0.01
	EG	113.05 ± 1.2 **	6.62 ± 0.16	20.22 ± 0.40	11.11 ± 0.35	116.01 ± 1.96 **	1566.25 ± 108.05	0.79 ± 0.05 *

Note: Data are presented as mean ± standard error. * Significant difference between the control group and the experimental group (*p* < 0.05). ** Highly significant difference between the control group and the experimental group (*p* < 0.01).

## Data Availability

The data that support the findings of this study are available from the corresponding author and the first author upon reasonable request.

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
