# Peer review of "Studies on the Phosphorus-Solubilizing Ability of Isaria cateinannulata and Its Influence on the Growth of Fagopyrum tataricum Plants"

_plants, 2024, doi:10.3390/plants13121694_

Round 1
Reviewer 1 Report
Comments and Suggestions for Authors
Dear authors, the manuscript "Exploration of the phosphorus-solubilizing ability of Isaria cateinannulata and its influence on the growth of Fagopyrum tataricum plants" present an interesting research in the field of phosphorus solubilizing natural resources. Some changes can act to improve the current work.
Abstract is clear and point the main findings of the study.
Introduction - this section has an appropriate length, introducing each of the background information needed for the topic, and presenting similar studies on other fungi. The aim and the objectives are presented in the last paragraph of the Introduction.
The Materials and method section present in detail each of the processes and techniques used by the authors. References to used formulas should be completed in the manuscript (e.g. lines 186, 187).
Results and Analysis - I suggest renaming this section as Results and Discussion.
Overall, this section presents the interpretation of the observations and data recorded by the authors, giving details related to differences and trends observed.
Some changes should be done in this section to improve its quality.
Table 1 need to be rewritten as in the journal template. The significant differences and the strains with the highest PSI should be detailed.
Figure 2 - the quality should be improved. A detailed explanation is given in the 3.2 sub-section.
Table 2 - same suggestion as for table 1. An extension of interpretation is needed.
Overall, this section is detailed and presents well the findings of this study.
Section 4 Discussion and Conclusion should be moved to the Results section. In this way, it will complete the details for the Results.
A conclusion section should to be created by the authors, to present the most important findings of the study and values for the highest parameters.
Author Response
Dear Assistant Editor Ms. Supakorn Nundaeng and Reviewer 1, Thank you for your letter and for the reviewers’ comments concerning our manuscript entitled “Studies on the Phosphorus-Solubilizing Ability of Isaria cateinannulata and Its Influence on the Growth of Fagopyrum tataricum Plants”(ID: plants-2994681). The comments are all valuable and very helpful for revising and improving our paper. We have studied comments carefully and have made correction which we hope meet with approval. Revised portion are marked in green in the paper. The main corrections in the paper and the responds to the editor’s and the reviewer’s comments are detailed in the attachment。 We have made several improvements to the manuscript.These changes do not influence the content and framework of the paper. We have marked all changes in the revised paper using different colors for clarity. . We sincerely appreciatethe Editors/Reviewers’ efforts and hope that the corrections will meet with approval. Once again, thank you very much for your comments and suggestions. Sincerely, Guimin Yang, Email: 222100100453@gznu.edu.cn; tel.: +86-18386103926; Xiaona Zhang, Associate Professor, Email: 201606003@gznu.edu.cn; tel.: +86-18286039295; fax: +86-0851-86780646.

Reviewer 2 Report
Comments and Suggestions for Authors
The paper entitled <Studies on the Phosphorus-Solubilizing Ability of 18 strains of I. cateinannulata : Isaria cateinannulata( Phosphate solubilizing microorganisms, Zygomycetes) on the Growth of Fagopyrum tataricum Plants> presents interesting results of an economic plant from China: F. tataricum. and microbial growth-promoting capacity to solubilize phosphorus , they test the solubilize phosphorus ability of 18 strains of I. cateinannulata by analyzing their growth in culture medium.
through field experiment They showed 18 strains of I. cateinannulata had phosphorus release capacity, with phosphorus solubilization, and one strain exhibited the best phosphorus solubilization effect. Additionally, the field results demonstrated that the micobe positively influenced the growth, root length, and yield of this plant.
The paper is well presented, however, hard to easily reading. Thus, I suggest to better explain the treatments in legends, including CK =.....
Please, check the repeatedsentence :.
Fig. 5: better explain if there are image of seeds?
line 11:
This elevation stems from the diminished need for phosphorus in coordinatio. This elevation stems from the diminished need for phosphorus in coordinating growth and development after ...
Fig, 3: legend: to indicate: CK = ....
EXPERIMENTAL GROUP = ....
--I suggest to better explore the capacity of controlling pests and diseases while promoting plant growth of genus Cordyceps (order Entomophthorales, and It serves as an entomopathogenic fungus found to enhance the growth of several crop plants, such as tomato through endophytic colonization.
Why authors present black and white images? please explain.
I highlight Figure 1, which illustrates the colony characteristics, that can serve for how to show results from other researchers.
In short, it is an Interesting paper, that could be better explained, useful for researchers and farmers.
line 117: 1000 mL beaker. Distilled /
please check, replace by: becker
line 115: 2.2. Preparation of Liquid Medium and Determination of Soluble Phosphorus Content of Strain/ better to use: strains.
Author Response
Dear Assistant Editor Ms. Supakorn Nundaeng and Reviewer 2, Thank you for your letter and for the reviewers’ comments concerning our manuscript entitled “Studies on the Phosphorus-Solubilizing Ability of Isaria cateinannulata and Its Influence on the Growth of Fagopyrum tataricum Plants”(ID: plants-2994681). The comments are all valuable and very helpful for revising and improving our paper. We have studied comments carefully and have made correction which we hope meet with approval. Revised portion are marked in green in the paper. The main corrections in the paper and the responds to the editor’s and the reviewer’s comments are detailed in the attachment。 We have made several improvements to the manuscript.These changes do not influence the content and framework of the paper. We have marked all changes in the revised paper using different colors for clarity. . We sincerely appreciatethe Editors/Reviewers’ efforts and hope that the corrections will meet with approval. Once again, thank you very much for your comments and suggestions. Sincerely, Guimin Yang, Email: 222100100453@gznu.edu.cn; tel.: +86-18386103926; Xiaona Zhang, Associate Professor, Email: 201606003@gznu.edu.cn; tel.: +86-18286039295; fax: +86-0851-86780646.

Round 2
Reviewer 1 Report
Comments and Suggestions for Authors
The authors have improved their work.